# 3p Arm Loss and Survival in Head and Neck Cancer: An Analysis of TCGA Dataset

**DOI:** 10.3390/cancers13215313

**Published:** 2021-10-22

**Authors:** Hugh Andrew Jinwook Kim, Mushfiq Hassan Shaikh, Mark Lee, Peter Y. F. Zeng, Alana Sorgini, Temitope Akintola, Xiaoxiao Deng, Laura Jarycki, Halema Khan, Danielle MacNeil, Mohammed Imran Khan, Adrian Mendez, John Yoo, Kevin Fung, Pencilla Lang, David A. Palma, Krupal Patel, Joe S. Mymryk, John W. Barrett, Paul C. Boutros, Luc G. T. Morris, Anthony C. Nichols

**Affiliations:** 1Department of Otolaryngology-Head and Neck Surgery, University of Western Ontario, London, ON N6A3K7, Canada; hkim2022@meds.uwo.ca (H.A.J.K.); mushfiq.shaikh@lhsc.on.ca (M.H.S.); yzeng2023@meds.uwo.ca (P.Y.F.Z.); asorgini@uwo.ca (A.S.); takintola2025@meds.uwo.ca (T.A.); xdeng2024@meds.uwo.ca (X.D.); ljarycki@uwo.ca (L.J.); Halema.Khan@lhsc.on.ca (H.K.); Danielle.Macneil@lhsc.on.ca (D.M.); mkhan953@uwo.ca (M.I.K.); Adrian.Mendez@lhsc.on.ca (A.M.); John.Yoo@lhsc.on.ca (J.Y.); Kevin.Fung@lhsc.on.ca (K.F.); david.palma@lhsc.on.ca (D.A.P.); jmymryk@uwo.ca (J.S.M.); john.barrett@lhsc.on.ca (J.W.B.); 2Memorial Sloan Kettering Cancer Center, Department of Surgery, New York, NY 10065, USA; mal2087@nyp.org (M.L.); morrisl@mskcc.org (L.G.T.M.); 3Department of Oncology, University of Western Ontario, London, ON N6A3K7, Canada; pencilla.lang@lhsc.on.ca; 4Moffitt Cancer Center, Department of Otolaryngology, Tampa, FL 33612, USA; krupal.patel@moffitt.org; 5Department of Microbiology & Immunology, University of Western Ontario, London, ON N6A3K7, Canada; 6Department of Human Genetics, University of California, Los Angeles, CA 90095, USA; pboutros@mednet.ucla.edu; 7Department of Urology, University of California, Los Angeles, CA 90095, USA; 8Eli and Edythe Broad Center of Regenerative Medicine and Stem Cell Research, University of California, Los Angeles, CA 90095, USA; 9Institute for Precision Health, University of California, Los Angeles, CA 90095, USA; 10Jonsson Comprehensive Cancer Centre, University of California, Los Angeles, CA 90095, USA

**Keywords:** head and neck cancer, chromosome loss, copy number alterations, genomics, mutational status

## Abstract

**Simple Summary:**

Loss of the 3p chromosome arm has previously been reported to be a predictor of poorer outcome in head and neck cancer regardless of human papillomavirus (HPV) infection. However, a useful measurement of 3p arm loss remains unclear, as well as its associations with survival and mutations in head and neck cancers. We found that HPV-negative tumors almost universally had close to all or no genes deleted on the 3p arm. 3p arm loss was not associated with survival regardless of HPV infection. HPV-negative tumors with 3p arm loss also had arm-level gene copy number changes in other chromosomes across the genome. Their extracellular environments had decreased immune cell activity and oxygen depletion. Abundances of two proteins were decreased and two micro-RNAs were increased, and these changes were associated with survival. Our findings suggest that 3p arm loss may not predict survival, but produces distinct biological characteristics in HPV-negative tumors.

**Abstract:**

Loss of the 3p chromosome arm has previously been reported to be a biomarker of poorer outcome in both human papillomavirus (HPV)-positive and HPV-negative head and neck cancer. However, the precise operational measurement of 3p arm loss is unclear and the mutational profile associated with the event has not been thoroughly characterized. We downloaded the clinical, single nucleotide variation (SNV), copy number aberration (CNA), RNA sequencing, and reverse phase protein assay (RPPA) data from The Cancer Genome Atlas (TCGA) and The Cancer Proteome Atlas HNSCC cohorts. Survival data and hypoxia scores were downloaded from published studies. In addition, we report the inclusion of an independent Memorial Sloan Kettering cohort. We assessed the frequency of loci deletions across the 3p arm separately in HPV-positive and -negative disease. We found that deletions on chromosome 3p were almost exclusively an all or none event in the HPV-negative cohort; patients either had <1% or >97% of the arm deleted. 3p arm loss, defined as >97% deletion in HPV-positive patients and >50% in HPV-negative patients, had no impact on survival (*p* > 0.05). However, HPV-negative tumors with 3p arm loss presented at a higher N-category and overall stage and developed more distant metastases (*p* < 0.05). They were enriched for SNVs in TP53, and depleted for point mutations in CASP8, HRAS, HLA-A, HUWE1, HLA-B, and COL22A1 (false discovery rate, FDR < 0.05). 3p arm loss was associated with CNAs across the whole genome (FDR < 0.1), and pathway analysis revealed low lymphoid–non-lymphoid cell interactions and cytokine signaling (FDR < 0.1). In the tumor microenvironment, 3p arm lost tumors had low immune cell infiltration (FDR < 0.1) and elevated hypoxia (FDR < 0.1). 3p arm lost tumors had lower abundance of proteins phospho-HER3 and ANXA1, and higher abundance of miRNAs hsa-miR-548k and hsa-miR-421, which were all associated with survival. There were no molecular differences by 3p arm status in HPV-positive patients, at least at our statistical power level. 3p arm loss is largely an all or none phenomenon in HPV-negative disease and does not predict poorer survival from the time of diagnosis in TCGA cohort. However, it produces tumors with distinct molecular characteristics and may represent a clinically useful biomarker to guide treatment decisions for HPV-negative patients.

## 1. Introduction

Cancer results from the stepwise accumulation of genetic mutation including chromosomal arm level copy number changes [1]. One of the most frequent genetic alterations in head and neck squamous cell carcinoma (HNSCC) is loss of the p arm of chromosome 3 [2,3]. In the Pan-Cancer Analysis of Whole Genomes (PCAWG) dataset [4], 3p loss is a preferentially early-clonal event in HNSCC occurring on average over 10 years prior to diagnosis. It generally follows mutation of NOTCH1, TP53, and TERT promoter in tumors with one or more of these derangements [5]. Indeed, 3p loss is found in 40% of oral dysplastic lesions and these lesions are clonally related to adjacent invasive HNSCC [6]. Moreover, dysplastic lesions carrying 3p loss are 33 times more likely to progress to invasive cancer than those without [7].

Importantly, two separate studies have identified 3p chromosome arm deletion as being enriched in treatment-resistant human papillomavirus (HPV)-negative and HPV-positive HNSCC [8,9]. Gross and colleagues recruited 250 TCGA and 48 University of Pittsburgh Medical Center (UPMC) HPV-negative samples and 52 TCGA and 7 UPMC HPV-positive samples [8]. They demonstrated markedly poorer survival for tumors with 3p loss in both cohorts, suggesting that it may be a clinically useful biomarker. Impressively, HPV-positive tumors without 3p loss had 100% survival at 3 years. In the study by Morris and colleagues, they utilized the Memorial Sloan Kettering integrated mutation profiling of actionable cancer targets (MSK-IMPACT) sequencing panel to characterize multiple histologies of recurrent head and neck cancer including 21 recurrent HPV-positive squamous cell cancers [9]. They identified that 3p arm loss was ninefold more frequent in HPV-positive recurrences versus primary tumors, further supporting its prognostic importance.

However, a clinically useful operational definition of 3p arm loss has yet to be clearly determined. Chromosome arm loss can be complete or partial and depends on the method of detection [3]. Potentially, these losses can be detected by exome sequencing, whole genome sequencing, targeted sequencing, or copy number microarrays [8,9]. Gross et al. defined the chromosomal status as the median copy number of 12 genes on the genetically unstable 3p14.2 locus (fragile site) derived from copy number microarrays [8]. Their justification for this method was that the majority of patients with 3p arm deletion had fragile site loss as well. However, a visual observation of their Appendix A shows that, while this definition is quite accurate for HPV-negative cases, HPV-positive 3p arm deletion was more closely associated with loss of the 3p21.1 locus. Morris et al. (2017) used the FACETS algorithm that derived 3p loss from the MSK Impact targeted sequencing panel [9,10]. A clearer definition of 3p loss is required to fully understand its prognostic and biologic importance.

In this paper, we define 3p arm deletion for both HPV-negative and HPV-positive tumors, and then use this definition in the complete The Cancer Genome Atlas (TCGA) dataset, comprised of nearly twice as many samples as previously studied, to examine the prognostic importance of this alteration. We reinforce our findings by correlating 3p status with survival in the independent cohort of both HPV-positive and -negative tumors, with 3p arm status defined using the MSK-IMPACT targeted sequencing panel. We then carry out a multi-omic analysis of tumors with and without 3p loss to build on previous studies, elucidating a wider spectrum of molecular differences between these two subsets of HNSCC tumors.

## 2. Materials and Methods

### 2.1. Study Populations

TCGA Level 3 DNA mutation packager calls data, copy number alterations (CNAs), non-normalized mRNA and miRNA sequencing calls, and merged clinical data sets were downloaded from The Broad Institute’s Firehose database’s most recent callsets (version GRCh38) and the study by Liu et al. [11,12]. TCGA cohort was all treated with primary surgery. Staging was based on the 8th edition of the American Joint Committee on Cancer staging system. Smoking history was defined as heavy for >20 pack-years and light for <20 pack-years. Each sample’s HPV status was assigned based on viral transcript detection [13]. To keep our methods consistent with Gross et al. [8], we focused our analysis on the 496 cases under the age of 85. The pathological stage was used for T-category, N-category, and overall stage.

As an independent secondary validation cohort, 245 patient samples with clinical and survival data from Memorial Sloan Kettering’s MSK-IMPACT HNSCC cohort were used. This cohort was limited to HNSCCs that were distantly metastatic and/or recurrent, and not all patients were necessarily treated with primary surgery. Additionally, a publicly available cohort from the Clinical Proteomic Tumor Analysis Consortium (CPTAC) containing 110 HPV-negative HNSCC patients was used for validation of RPPA data.

### 2.2. Determining 3p Arm Status

In TCGA cohort, for HPV-negative samples, 3p arm deletion was defined as loss of ≥97% of chromosomal material from one arm. In contrast, patients with wildtype 3p arm exhibited less than 1% loss of the genes on the 3p arm. In HPV-positive samples, the thresholds were over 50% and less than 1%. Samples between the high and low thresholds were excluded. Determination of 3p arm status by the fragile site definition was done using the method defined by Gross et al. [8]. In the MSK-IMPACT validation cohort, full 3p arm loss was defined as >80% of the 3p arm genes deleted, and any 3p arm loss as any gene on 3p arm deleted.

### 2.3. Statistics

#### 2.3.1. Clinical Features

All statistical analyses were carried out in the R environment (version 3.6.1). Fisher’s exact test, Pearson’s χ^2^ goodness of fit test, and Mann–Whitney U tests were used to compare clinical features by 3p arm status in HPV-positive and -negative groups.

#### 2.3.2. Survival Analysis

The survival package (v 2.44-1.1) [14] was used for survival analyses. In both TCGA study and MSK-IMPACT validation cohorts, overall survival outcomes were compared using a log-rank test by 3p arm status in an independent HPV-positive and HPV-negative patient subsets, and Kaplan–Meier curves were constructed. Cox proportional hazards models with Wald *p*-values were used to conduct multivariate survival analyses in TCGA cohort with clinical covariates including 3p arm status and adjuvant radiotherapy.

#### 2.3.3. Exome Sequence Mutations

In TCGA cohort, the maftools package (version 2.0.10) [15] within the bioconductor framework was used for exome sequencing analyses. Mann–Whitney U test was used to compare total numbers of SNV mutations between tumors with and without 3p arm loss, and two-tailed *p* values were reported. Fisher’s exact tests were used to discover differentially mutated genes by 3p arm status. Synonymous mutation variants and mutations of the TTN gene were excluded from the analysis because TTN carries a high frequency of passenger mutations [16]. Furthermore, only genes mutated in at least ten patients in at least one comparison arm were evaluated. The obtained *p*-values were corrected for false discovery rates (FDRs) using the Benjamini–Hochberg method, and the FDR threshold for significance was set at 0.1, as we have done in the past [17,18,19].

#### 2.3.4. Copy Number Alterations

In TCGA cohort, individual gene frequencies of deep and shallow deletions as well as gains and amplifications, as termed by GISTIC2 analysis of the HNSCC cohort, were compared between tumors with and without 3p arm loss. Deep deletions (GISTIC2 value: −2) were defined as homozygous losses, and shallow deletions (GISTIC2 value: −1) as heterozygous losses. Gains (GISTIC2 value +1) and amplifications (GISTIC2 value +2) were characterized by differing degrees of copy number increases. Fisher’s exact tests were used to compare the CNA frequencies by 3p arm status, and FDR correction was performed as previously described [17,18,19].

#### 2.3.5. mRNA Sequencing Counts

In TCGA cohort, the DESeq2 package (version 1.24.0) [20] within the bioconductor framework was used to normalize and analyze TCGA HNSCC mRNA sequencing (mRNA-seq) count data. The mRNA abundance profiles of tumors with and without 3p arm loss were fit to negative binomial generalized linear distributions and compared by Wald tests with log_2_ fold-change shrinkage. FDR correction was performed as previously described [17,18,19]. The FDR threshold for significance was set at a stricter 0.01, given the size of the mRNA data interrogated.

#### 2.3.6. Integration of CNA and mRNA-Seq Results

Copy number gains and deletions corresponding to higher and lower mRNA abundance, respectively, are more likely to be biologically relevant [21,22]. With this reasoning, we plotted the log10FDR values from our CNA analysis against those from our mRNA-seq analysis. Genes were first filtered for FDR significance less than 0.1 and 0.01 from the CNA and mRNA comparisons, respectively, and at least a twofold change in mRNA abundance level. These genes, combined with our list of significant SNV findings, were analyzed for pathway over-representation using the Reactome Pathway Analysis [23] within the bioconductor framework.

#### 2.3.7. Tumor Microevironment Estimation

In TCGA cohort, the immunedeconv package (v 2.0.0) implementing the microenvironment cell populations-counter (MCP-counter) method [24] was used to estimate the cellular composition scores of each tumor’s microenvironment. These scores are based on validated transcriptomic markers known to specifically characterize the abundance of each specific immune cell population within the tumor. The method has been shown to be highly accurate compared with other known immune deconvolution methods, and capable of inter-sample comparisons [25]. Scores for T cells, natural killer (NK) cells, B cells, monocytes, myeloid dendritic cells (MDCs), neutrophils, endothelial cells, and cancer-associated fibroblasts (CAFs) were normalized using the Box–Cox transformation and compared using linear regression. Univariable regression followed by multivariable regression controlling for PIK3CA gain was used. We chose PIK3CA to independently control for the 3q26-29 amplicon, which has recognized associations with oncogenic pathways and, most significantly, with 3p arm loss in our and previous studies [26,27]. FDR correction was performed as previously described [17,18,19].

#### 2.3.8. Hypoxia Estimation

Hypoxia scores from eight validated gene signatures were downloaded from Bhandari et al.’s Appendix A [28]. In TCGA cohort, scores were summed up by assigning +1 to the top 50% of patients and −1 to the bottom 50% of patients based on mRNA abundance for each gene of each signature. Scores were normalized using the Box–Cox transformation and compared using linear regression. Univariable regression followed by multivariable regression controlling for PIK3CA gain was used. FDR correction was performed as previously described [17,18,19].

#### 2.3.9. Reverse Phase Protein Array

Level 3 processed normalized RPPA data were retrieved from the Cancer Proteome Atlas [29,30]. In TCGA cohort, protein abundances were normalized using the Box–Cox transformation and compared using linear regression. Univariable regression followed by multivariable regression controlling for PIK3CA gain was used. FDR correction was performed as previously described [17,18,19]. The differentially abundant proteins were validated with their mRNA abundance values using Spearman’s rank correlation (rho > 0, FDR < 0.1). Cox proportional hazards modeling with Wald *p*-values was performed in a dichotomizing manner by protein expression level above and below the median.

The CPTAC cohort had data on disease-free survival, but not overall survival. Cox proportional hazards modeling of disease-free survival with Wald *p*-values was performed in a dichotomizing manner by protein expression levels of ANXA1 and HER3 levels above and below the median.

#### 2.3.10. miRNA Sequencing Analysis

The DESeq2 package (version 1.24.0) [20] within the bioconductor framework was used to normalize and analyze TCGA HNSCC mRNA sequencing (mRNA-seq) count data. The mRNA abundance profiles of tumors with and without 3p arm loss were fit to negative binomial generalized linear distributions and compared by Wald tests with log_2_ fold-change shrinkage. FDR correction was performed as previously described [17,18,19]. The multiMiR package (version 1.8.0) [31] was used to search for miRNA targets. The differentially expressed miRNAs were then validated against target mRNA abundance values using Spearman’s rank correlation (FDR < 0.1). Cox proportional hazards modeling with Wald *p*-values was performed in a dichotomizing manner by miRNA abundance above and below the median.

## 3. Results

### 3.1. Definitions and Cohort Characteristics

#### 3.1.1. Defining the Study and Validation Cohorts

Advanced age and HPV infection produce tumors with distinct clinical and molecular characteristics [32]. Thus, we used an approach similar to that of Gross et al. [8], who analyzed 250 HPV-negative and 52 HPV-positive TCGA samples, based on an age cutoff of 85. This limited our analyses to the 496 patients in the expanded TCGA dataset under the age of 85 with clinical and copy number alteration (CNA) data. HPV status was determined by the detection of viral transcripts as determined by Bratman et al. [13], which resulted in 423 HPV-negative and 73 HPV-positive tumors (419 and 73 samples, respectively, with DNA mutation data and survival information).

A second cohort based on patient samples with clinical and survival data from 245 HNSCC patients from Memorial Sloan Kettering with tumors characterized with the MSK-IMPACT targeting sequencing panel was also analyzed. In this cohort, there were 94 HPV-negative and 77 HPV-positive tumors that had 3p arm status defined (Appendix A).

#### 3.1.2. Defining a Threshold for 3p Arm Deletion Status

Our definition of 3p arm loss included the chromosomal material between 3p11.1 and 3p26.3 inclusive, while maintaining the integrity of the telomere and centromere regions. Homozygous deletion of a chromosome arm is a distinct biological phenomenon [33]. In our cohort, homozygous deletion of the entire 3p arm was rare (only 4/496 tumors had homozygous deletion of greater than 50% of the 3p arm, Appendix A), thus these patients were excluded. For the remaining 492 samples, we determined the percentage of the 642 genes on the 3p arm that were deleted from a chromosome for each sample. Through this analysis, we determined a threshold for 3p deletion in both the HPV-positive and -negative cohorts. Our threshold resulted in deletion and wild-type cohorts that were clinically distinct without excluding a large number of samples. The distributions of patients with different percentages of the 3p arm deleted in the HPV-positive and -negative cohorts were plotted in Figure 1. We observed that most of the HPV-negative patients had almost complete arm loss (>97% genes deleted) or almost complete arm preservation (<1% genes deleted). Using the >97% and <1% thresholds to define our HPV-negative cohorts resulted in 293 patients in the 3p arm lost cohort and 82 in the 3p arm preserved cohort, and 48 excluded with between 1% and 97% loss. The HPV-positive patients also showed an approximate bimodal distribution, but the proportion of patients with between 50% and 90% deletion was too great to exclude. Thus, using the >50% and <1% thresholds to define our HPV-positive samples resulted in 28 HPV-positive patients in the 3p arm lost cohort and 40 in the 3p arm preserved cohort, with 5 excluded with between 1% and 50% loss.

We compared our results to the fragile site definition of 3p arm deletion used by Gross and colleagues [8]. In HPV-negative samples, 90% of the patients with fragile site loss also had 97% or more of the 3p arm deleted, while only 1% without fragile site loss did (*p* < 10^−64^, Appendix A). In HPV-positive samples, 90% of the patients with fragile site loss also had 50% or more of the 3p arm deleted, while 0% without fragile site loss did (*p* < 10^−16^, Appendix A). Tumors with 3p14.2 fragile site loss had a lower proportion of HPV infection than those without fragile site loss (8% vs. 31%, *p* < 10^−9^, Appendix A).

#### 3.1.3. Clinical Characteristics Including Stage at Presentation and Distant Metastasis Differ by 3p Deletion Status

We analyzed TCGA HNSCC cohort to determine whether our threshold definition of 3p deletion status was associated with any clinical or epidemiologic features. In the HPV-negative cohort, patients with 3p arm loss compared with patients without 3p arm loss were more likely to be male (76% vs. 51%, *p* < 10^−4^), to have laryngeal primaries (28% vs. 10%, *p* < 10^−3^), and to have heavier smoking histories (64% vs. 39%, *p* < 10^−3^, Table 1). They also presented with advanced nodal disease (N2b-N3, 39% vs. 19%, *p* = 0.0018) and overall stage (stage IV, 63% vs. 49%, *p* = 0.033), and developed more distant metastases on follow-up (5% vs. 0%, *p* = 0.049, Table 1). In the HPV-positive cohort, 3p arm loss was associated with a heavier smoking history (61% vs. 24%, *p* = 0.015) (Table 2). No significant differences were seen with other demographic variables between these cohorts, at least at this level of statistical power.

### 3.2. Survival Outcomes by 3p Arm Deletion Status

#### 3.2.1. Overall Survival Outcomes Were Not Associated with 3p Arm Deletion Status

Kaplan–Meier curves were generated comparing overall survival by 3p deletion status, measured first using Gross et al.’s fragile site definition and then our threshold definition (Figure 2). There was a trend towards poorer survival for patients with the 3p fragile site lost versus preserved in HPV-negative tumors (HR = 1.4, 95% CI = 0.96–2.07, *p* = 0.079, Figure 2A). This survival difference remained only a trend towards significance on multivariate analysis (*p* = 0.067, Appendix A). There was no difference in overall survival between fragile site lost and preserved tumors in the HPV-positive cohort (Figure 2B). There were no survival differences between 3p arm lost and preserved tumors by the threshold definition in HPV-positive or -negative cohorts on univariate or multivariate analysis (Figure 2C,D, Appendix A).

#### 3.2.2. External Validation of Overall Survival Outcomes

Using the independent MSK-IMPACT validation cohort, there were again no differences in overall survival associated with full 3p arm loss in HPV-negative (*p* = 0.49) or HPV-positive (0.42) samples, and the same was true for any 3p arm loss in HPV-negative (*p* = 0.32) or HPV-positive (0.23) samples (Appendix A).

### 3.3. Mutational Differences by 3p Arm Deletion Status

#### 3.3.1. SNV Mutations Differ Significantly by 3p Arm Deletion Status

The total somatic mutation load of single nucleotide variations (SNVs) did not differ between 3p arm lost and preserved groups in the HPV-negative (*p* = 0.098) or -positive (*p* = 0.66) cohorts (Appendix A). The somatic non-synonymous SNV profiles of the HPV-negative HNSCC cohort were compared by 3p deletion status, measured using our threshold definition. Seven genes had significantly different non-synonymous SNV frequencies after multiple-testing correction: CASP8, HRAS, TP53, HLA-A, HUWE1, HLA-B, and COL22A1 (FDR < 0.1, Figure 3). There were no genes with differential frequencies of SNVs by 3p arm status within the HPV-positive cohort.

#### 3.3.2. Copy Number Aberrations Are Widespread in 3p Deletion

Analogous to mutation load of SNVs, percent genome altered (PGA) measures the burden of CNAs across the tumor genome. Although there was no difference in mutation load, 3p arm lost tumors had greater PGA than 3p arm preserved tumors in the HPV-negative cohort (*p* < 10^−36^, Appendix A). Interestingly, in tumors with 3p arm lost, HPV-negative tumors had greater PGA than HPV-positive tumors (*p* < 10^−9^), but in tumors with 3p arm preserved, the opposite was observed, with greater PGA in HPV-positive tumors than in HPV-negative tumors (*p* < 10^−5^) (Appendix A).

We compared CNA differences between HPV-negative tumors with and without 3p arm deletion using our threshold definition. As expected, the 642 genes on the 3p arm were deleted at a higher frequency in tumors with 3p arm loss. Not counting the genes on the 3p arm, we found 20,515 genes with more shallow deletions, 1 gene with fewer shallow deletions, 15,506 genes with more gains, and 170 genes with fewer gains in tumors with 3p arm loss (FDR < 0.1, Appendix A). These CNA events spanned all 24,134 genes (not counting 3p arm genes) in the human genome sequenced by TCGA, with an overlap of 12,058 genes that had gains and deletions at varying frequencies between tumors with and without 3p arm loss (Appendix A). Two genes, CDH13 (FDR < 10^−4^) and NCKAP5 (FDR = 0.0025), had fewer deep deletions and 458 genes had more amplifications in tumors with 3p arm loss (FDR < 0.1, Appendix A).

The CNA differences, in terms of both shallow deletions and amplifications, were prevalent throughout most of the chromosome arms and associated with strong significance values (Figure 4). There was a marked association of 3q arm gains (top 730 CNAs ranked by significance, FDR < 10^−15^) and amplifications (420/458 significant amplifications, FDR < 0.1) occurring in conjunction with 3p arm loss. There were no CNA differences found between HPV-positive tumors with and without 3p arm loss; however, this may be owing to limited statistical power given the small sample size of HPV-positive tumors.

#### 3.3.3. Pathway Analysis of CNAs Integrated with mRNA Data

Given that CNAs with biological relevance are often associated with a corresponding change in mRNA transcript levels [34], we filtered our large list of CNA results through mRNA abundance data in the HPV-negative cohort. Analysis of the mRNA data revealed 796 genes that had significantly higher abundance in tumors with 3p arm deletion and 307 genes (including genes of the 3p arm) with lower abundance in tumors with 3p arm deletion (FDR < 0.01, absolute log_2_ fold change > 1, Appendix A). Because we identified many genes with significant CNA differences in both deletion and amplification analyses, we based those genes’ copy number status on the analysis with a lower FDR value. We found 309 genes with a higher copy number and mRNA abundance, and 162 genes (excluding genes of the 3p arm) with a lower copy number and mRNA abundance in tumors with 3p arm deletion (Appendix A, Appendix A). Of the genes on the 3p arm, 12 had lower mRNA abundance (FDR < 0.01, absolute log fold change > 1) in addition to lower copy number in tumors with threshold arm loss: RPL32, EOMES, ACAA1, CXCR6, CCR5, UBA7, RPL29, CCR2, ALS2CL, PLCD1, DNASE1L3, and XCR1.

Pathway analysis was performed on the combined list of SNVs (*n* = 7) and the subset of CNAs filtered with mRNA-seq data (*n* = 483) that differed significantly in tumors with 3p arm loss. The enriched pathways included upregulation of neuronal signalling in 3p loss tumors, and downregulation of immunoregulatory interactions between lymphoid and non-lymphoid cells and chemokine receptor binding (FDR < 0.1, Appendix A).

#### 3.3.4. Comparison of Tumor Microenvironments

The tumor microenvironment (TME) modulates tumor growth and impacts treatment response [35]. We analyzed the composition differences between tumors with and without 3p arm loss by deconvolution of the mRNA abundance data. Within the HPV-negative cohort, cancer-associated fibroblast (CAF) markers were more abundant in 3p deletion, while T cell, NK cell, and monocyte markers were more abundant in tumors without 3p deletion (FDR < 0.1, Figure 5, Appendix A). Even when controlling for PIK3CA gain, T cell, NK cell, and monocyte markers were more abundant in tumors without 3p deletion (FDR < 0.1, Appendix A). There were no TME differences found by 3p arm status within the HPV-positive cohort.

In addition to stromal cells, varying hypoxic conditions in the TME predicted clinical prognosis and treatment resistance [28,36]. We compared eight hypoxia scores between tumors with 3p arm loss and those without in the HPV-negative cohort. We found consistently higher hypoxia levels in tumors with 3p arm loss across all hypoxia scores (FDR < 0.1, Figure 6, Appendix A). Even when controlling for *PIK3CA* gain, four of the eight hypoxia scores were significantly higher in tumors with 3p arm loss (Appendix A). There were no differences in hypoxia profiles within the HPV-positive cohort.

#### 3.3.5. HER3-pY1289 and ANXA1 Proteins Were Less Abundant in 3p Lost Tumors and Were Associated with Survival Outcomes

We analyzed the RPPA results from The Cancer Protein Atlas (TCPA) [29,30] and found 59 differentially abundant proteins and phosphoproteins by 3p deletion status (FDR < 0.1, Appendix A). Twenty-six of these were more highly abundant in tumors with 3p arm loss, and 33 were more highly abundant in tumors without it. Protein levels often have a poor correlation with mRNA levels owing to complex regulatory mechanisms, but positive correlation is more likely to be associated with biological relevance [37]. Thirty-three of the 59 differentially expressed proteins, including ANXA1 (*p* = 0.68, *p*~0), had significant positive correlations of their levels with mRNA (FDR < 0.1, Appendix A). Sixteen of these were more highly expressed in tumors with 3p arm loss, and 17 were more highly expressed in tumors without it.

On controlling for *PIK3CA* gene gain, 26 proteins and phosphoproteins were differentially abundant, including 12 more abundant in tumors with 3p arm loss and 14 in tumors without (Appendix A). PI3K-p110-α, which had increased abundance in tumors with 3p arm loss in univariable analysis (fold change = 1.5, 95% CI = 1.1–2.0 FDR = 0.042), had significantly increased abundance with PIK3CA gain (fold change = 2.1, 95% CI = 1.5–2.9, FDR = 0.0044), but not with 3p arm loss in multivariable analysis.

Furthermore, two of the differentially expressed proteins, with and without control for PIK3CA gene gain, were associated with survival. HER3-pY1289 abundance was lower in tumors with 3p arm loss, and this was associated with favorable overall survival (HR = 0.52, 95% CI = 0.35–0.76, FDR = 0.039). Conversely, ANXA1 also had lower abundance in tumors with 3p arm loss, but this was associated with poorer overall survival (HR = 1.80, 95% CI = 1.23–2.63, FDR = 0.071) (Figure 7A,B, Appendix A). Using our CPTAC validation cohort, low abundance of ANXA1 trended towards poorer disease-free survival (HR = 2.0, *p* = 0.085, Appendix A), but not HER3.

#### 3.3.6. hsa-miR-548k and hsa-miR-421 Were More Abundant in 3p Arm Loss Tumors and Were Associated with Poor Survival

Comparison of the miRNA transcript levels revealed 201 miRNAs differentially abundant between tumors with and without 3p arm loss. Thirty of these had validated mRNA targets as searched by multiMiR [31], and 136 of these mRNA targets had significant abundance correlation with the targeting miRNA (FDR < 0.1, Appendix A). Two miRNAs, hsa-miR-548k (HR = 1.7, FDR = 0.047) and hsa-miR-421 (HR = 1.6, FDR = 0.054), had higher expression in 3p arm loss associated with poorer overall survival (Figure 7C,D, Appendix A).

## 4. Discussion

In this study, we have examined different ways of defining 3p arm loss and surveyed the biological differences associated with this genomic event. We first established an alternative definition of 3p arm loss as a primarily all-or-none event in the HPV-negative cohort. Applying controls for PIK3CA gain status, we found that HPV-negative tumors with 3p arm loss are molecularly distinct in terms of SNVs CNAs, mRNA and protein abundance, and TME compared with HPV-negative tumors with intact 3p arms. Our findings are much more expansive than previously conducted research on 3p arm loss as they encompass more patient data and types of analysis than previously undertaken. Furthermore, we have re-examined the association of 3p arm loss with survival using the aid of an external MSK-IMPACT cohort. Regardless of the definition of 3p arm loss, we were unable to confirm, either with TCGA or MSK-IMPACT validation cohorts, an association with survival in either HPV-positive or negative HNSCC, as has been previously reported [8,9]. However, a subset of the protein and miRNA differences identified were associated with survival in HPV-negative disease.

The lack of association with survival was surprising, particularly as the study by Gross and colleagues used an earlier subset of TCGA dataset before data from the full cohort were released, along with additional samples from the University of Pittsburgh Medical Centre (UMPC) [8]. Despite significant overlaps in their study population and ours, and use of the identical fragile site definition, we failed to identify the same survival differences in our interrogation of TCGA cohort or the MSK-IMPACT validation cohort. One possible source for this discrepancy is how HPV-positive tumors were defined. In the Gross et al. study, most HPV calls were made from sequencing data from TCGA HNSCC analysis working group; however that analysis was not completed for all samples. Thus, HPV status was supplemented with PCR-based MassARRAY HPV analysis. In our study, HPV status was based solely on the detection of HPV transcripts from the RNA sequencing, as determined by Bratman et al. [13]. It is important to know that TCGA was not primarily designed to collect survival outcomes, and shortcomings in the outcomes have been identified and “cleaned up” by TCGA study group in 2018, several years after the Gross study was published [12]. We have used the most recent dataset provided in this update; however, careful curation of the treatment regimens by others has revealed that 30% of patients with TCGA received treatments that did not conform to the NCCN guidelines from twenty different institutions [38]. Nevertheless, our analysis suggests that 3p arm status may not be a useful biomarker of survival in HPV-positive or negative HNSCC.

Although tumors with or without 3p arm loss had no differences in survival, they were molecularly distinct, and some of these molecular differences were associated with survival. Our analysis identified two proteins, ANXA1 and HER3, that had significantly lower abundance in tumors with 3p arm loss after controlling for PIK3CA gain, and were associated with survival outcomes in HPV-negative tumors. Low abundance of ANXA1, seen in tumors with 3p arm loss, was associated with poor overall survival, and this trend was also observed in our CPTAC validation cohort. This may have a mechanistic underpinning, as knockdown of ANXA1 has been shown to promote radio-resistance by reducing the levels of reactive oxygen species and promoting DNA repair in HNSCC [39]. The tumor suppressor protein P53 normally upregulates transcription of ANXA1 [39], so the nearly ubiquitous early initiating TP53-3p co-mutation event observed in the present study (93% of patients had TP53 mutations) and previous studies [5,8] may lead to this ANXA1 knockdown phenotype and associated treatment resistance.

In contrast to ANXA1, a lower abundance of activated HER3 (pY1289) was associated with improved survival in HPV-negative patients. The phosphorylation of HER3 at Y1289 is a PI3K binding site and is essential for PI3K/AKT pathway activation [40]. While phospho-HER3 had lower abundance levels, other PI3K/AKT pathway members PI3K-p110-α and HER2 were more abundant in the 3p arm loss group, only when not controlling for PIK3CA gain. This is not surprising, given that the p110 subunit of PI3K is encoded by the PIK3CA gene [26]. ERBB2 amplification and over-expression in HNSCC confer treatment resistance to cetuximab [41], and its protein product HER2 normally dimerizes with HER3 to bind PI3K and activate the pathway [42]. However, in breast cancers with concurrent low HER3 abundance and ERBB2 amplification, HER2 signalling is maintained by homodimerization. Loss of HER3-mediated resistance renders this subset particularly susceptible to HER2-targeting trastuzumab [43], which may be a potential therapeutic strategy for HNSCC with 3p arm loss and PIK3CA gain.

At the miRNA level, we found that high levels of hsa-miR-548k and hsa-miR-421 predicted poor overall survival in tumors with 3p arm loss. The effect of hsa-miR-538k on decreased survival among HNSCC with concomitant 3p arm loss and *TP53* mutation has been described by Gross et al., although, similarly to our study, no conclusions could be made on a causative relationship [8]. Hsa-miR-548k has also been identified as one component of the seven-miRNA panel crafted by Lu et al. that predicted poor overall survival in HNSCC [44]. A study in esophageal cancer found hsa-miR-548k to enhance malignant phenotypes and tumor progression [45].

Hsa-miR-421 has been suggested to induce cell growth and apoptosis resistance in HNSCC through inactivation of FOXO4 [46]. Numerous studies have identified hsa-miR-421 as a prognostic and diagnostic biomarker in many cancers; for example, high abundance is associated with poor prognosis in gastric cancer and osteosarcoma [47,48]. Larger prospective datasets are necessary to validate the prognostic importance of these miRNA.

Several cell cycle proteins had differential abundance when stratified by 3p arm status, including higher CHK2, CDK1-pY15, and RB in the 3p arm loss group and higher CHK1-pS296 and CYCLINE1 in the 3p arm preserved group. In response to DNA damage, phosphorylated CHK proteins maintain inhibitory phosphorylation of CDK proteins to block cell cycle progression at the G1/S and G2/M checkpoints [49]. Low levels of inactive CDK1-pY15 in the context of high CHK1-pS296 activity may be explained by a CDK1-addicted phenotype insensitive to inhibition by CHK1 in tumors with 3p arm preserved [50]. RB normally maintains the cell in G1, but, in HNSCC, it is one of the first mechanisms to fail via phosphorylation and inactivation by CYCLINE1 complexed with CDK2 [32,51]. Taken together, these protein abundance patterns are suggestive of cell cycle deregulation at several critical transition checkpoints in tumors with 3p arm preserved. This subset of patients may benefit from cell cycle inhibitors, specifically CHK inhibitors [50].

The most dramatic CNA events we observed were large gains and amplifications of the 3q arm. Gains and amplifications of the 3q arm occurring concomitantly with 3p arm loss have been reported in previous studies [27], and one possible explanation is formation of a 3q isochromosome [52]. Notably, 3q arm gains and amplifications in HPV-negative HNSCCs contribute to poor prognosis through increased oncogene expression [53,54]. In addition to known oncogenes such as *PIK3CA* and *TERT*, ten genes on the 3q arm have been identified to independently and synergistically contribute to poor survival when amplified [26]. Future studies will be needed to fully understand the biological implications of 3q arm gains and amplifications, particularly with concomitant 3p loss.

In the TME, tumors with 3p arm loss had low levels of T cells, NK cells, and monocytes after controlling for PIK3CA gain, and pathway analysis demonstrated impaired interactions between lymphoid and non-lymphoid cells and chemokine signalling. Combined, these suggest an underactive cell-mediated immune response in tumors with 3p arm loss, which has previously been linked to poor survival and treatment resistance in HNSCCs [55]. Specifically, the loss of chemokine receptor genes enriched in the 3p arm likely contributes to impaired lymphocyte migration and anti-PD-L1 resistance [56,57]. According to Ribas et al., the absence of T cell infiltration precludes the need for tumor cells to express PD-L1, and indeed we observed a lower expression in tumors with 3p arm loss as well. Similarly, a previous study reported that 9p arm-level loss was strongly predictive of T cell and immune score depletion, and JAK2-PD-L1 codeletion at 9p24 predicted poor survival after anti-PD-1 therapy in HPV-negative HNSCCs [58]. These T cell- and PD-L1-double-negative aneuploid tumors may be best treated with a combination of anti-PD-L1 nivolumab with anti-CTLA-4 ipilimumab. Ipilimumab diversifies T cell receptors and increases T cell infiltration, allowing tumor cells to reactively express PD-L1, which is simultaneously blocked by nivolumab [59]. This combination therapy compared with nivolumab alone has demonstrated efficacy in extending progression-free survival in patients with metastatic melanoma [60].

Moreover, in the TME, we unanimously found higher levels of hypoxia in tumors with 3p arm loss using scores defined by four different groups after controlling for PIK3CA gain [28]. Hypoxia is known to increase genomic instability and contribute to poor survival outcomes, metastasis, and chemoradioresistance [28], which could explain the mutational and clinical patterns observed with 3p arm loss cohorts in other studies [8,9]. Hypoxia-activated pro-drugs, in development for HNSCC treatment, may mitigate the treatment-resistant phenotype of 3p arm loss and improve clinical outcomes [36].

## 5. Conclusions

In summary, we have defined and performed analyses with a new threshold-based classification of 3p arm loss tumors in the computational research setting, including control for PIK3CA gain and validation cohorts. Biological differences were found at the exome, chromosome, transcriptome, proteome, and TME levels between tumors dichotomized using our definition. Future studies are needed to investigate if 3p arm loss has a causative role, and to conclusively determine the prognostic importance of 3p arm loss as a biomarker of survival.

## Figures and Tables

**Figure 1 cancers-13-05313-f001:**
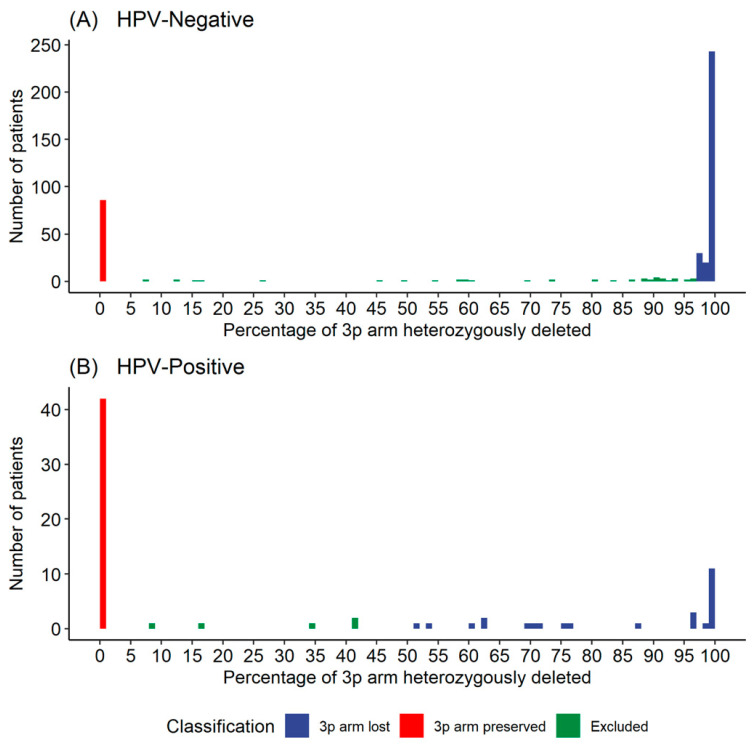
Number of patients by threshold. (**A**) Human papillomavirus (HPV)-negative samples, (**B**) HPV-positive samples. Thresholds for cohort selection set at 97% high and 1% low for HPV-negative samples, and 50% high and 1% low for HPV-positive samples.

**Figure 2 cancers-13-05313-f002:**
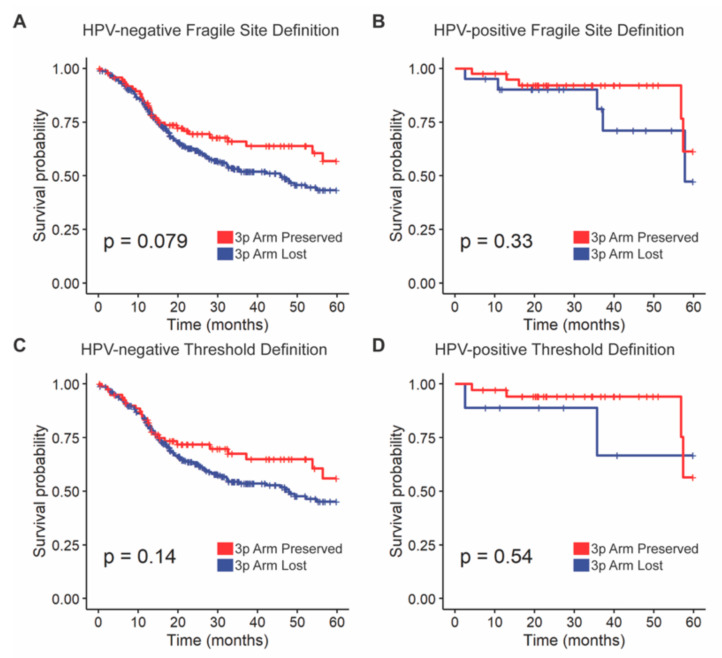
Overall survival stratified by 3p deletion status. (**A**) Fragile site definition in HPV-negative samples, (**B**) fragile site definition in HPV-positive samples, (**C**) threshold definition in HPV-negative samples, and (**D**) threshold definition in HPV-positive samples. Log-rank tests were used for survival comparisons.

**Figure 3 cancers-13-05313-f003:**
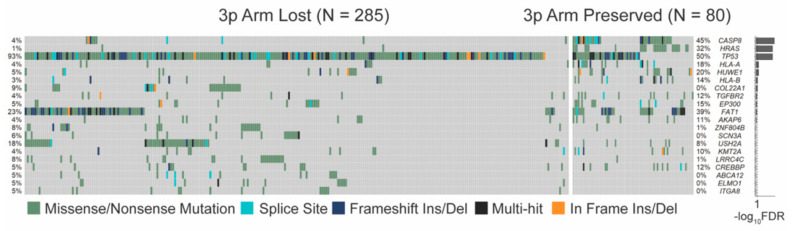
Top 20 single nucleotide variation (SNV) differences between threshold arm loss and no loss of the 3p arm in HPV-negative samples. CASP8, HRAS, TP53, HLA-A, HUWE1, HLA-B, and COL22A1 met significance (false discovery rate (FDR) < 0.1).

**Figure 4 cancers-13-05313-f004:**
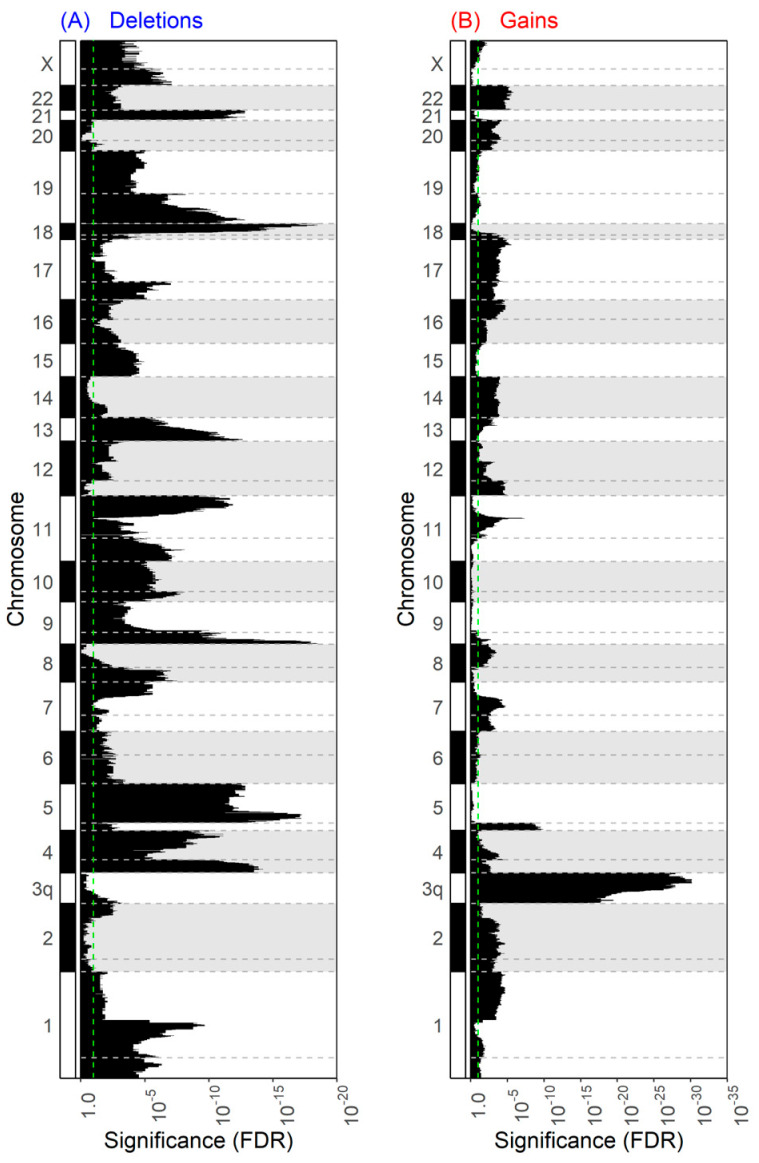
Chromosomal mapping of shallow deletions and gains by 3p arm status. (**A**) Shallow deletion and (**B**) gain frequency were compared between tumors with and without 3p arm loss using Fisher’s exact test. The FDR significance values of these comparisons, corrected through the Benjamini–Hochberg method, were plotted by chromosomal mapping location. Genes on the 3p arm are not shown. The green dotted line marks the significance threshold of 0.1.

**Figure 5 cancers-13-05313-f005:**
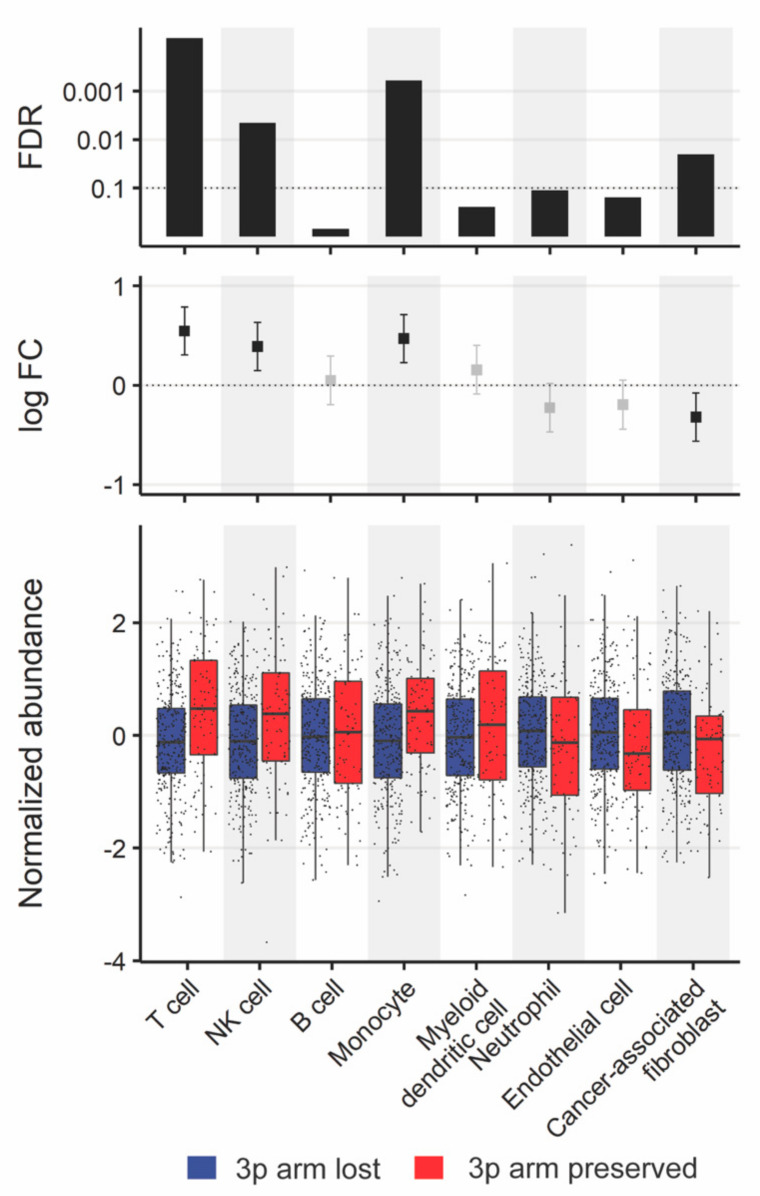
Tumor microenvironment (TME) differences by 3p arm status. Box plots were constructed with overlaid dot plots of MCP-counter scores of each TME cell type normalized using the Box–Cox transformation. Mann–Whitney U test was used for comparison. The 95% confidence intervals of the log fold changes (FC) colored black for significant and grey for insignificant changes, as well as significant FDR values (<0.1) corrected through the Benjamini–Hochberg method, are shown.

**Figure 6 cancers-13-05313-f006:**
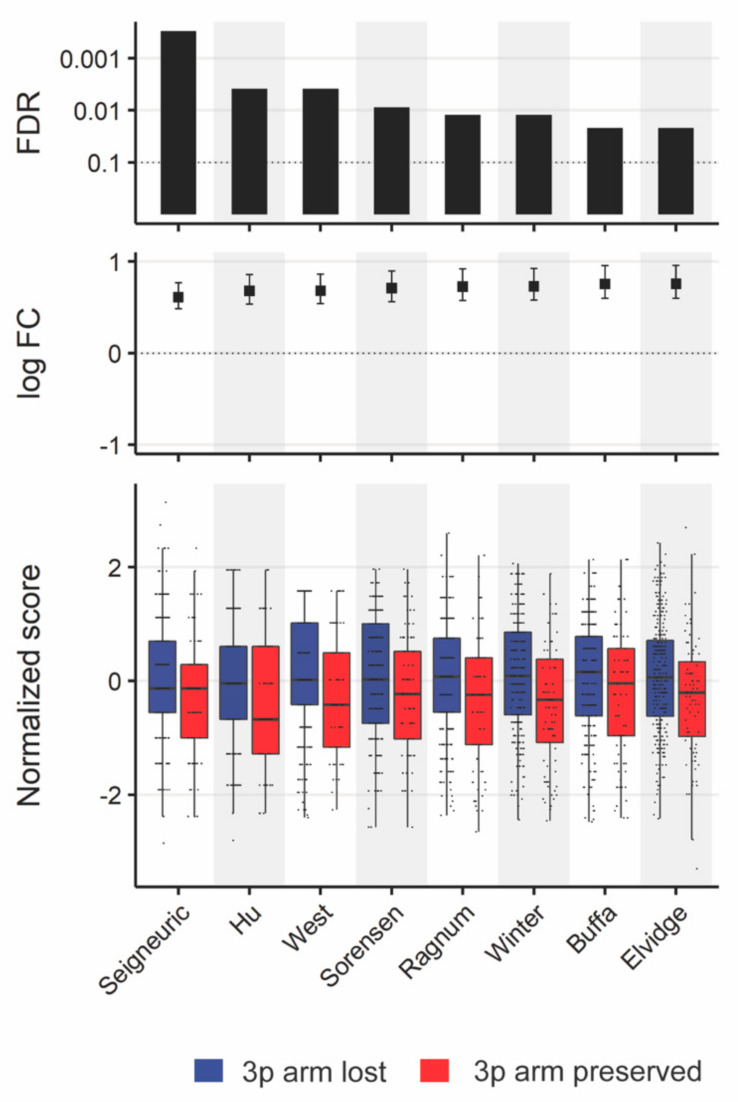
Hypoxia differences by 3p arm status. All eight hypoxia scores in Bhandari et al.’s Appendix A [28] were normalized using Box–Cox transformation and compared using linear regression. The 95% confidence intervals of the log fold changes are colored black for significant and grey for insignificant changes, and significant FDR values corrected through the Benjamini–Hochberg method are shown. In all eight comparisons, tumors with 3p arm loss had significantly higher hypoxia scores (FDR < 0.1).

**Figure 7 cancers-13-05313-f007:**
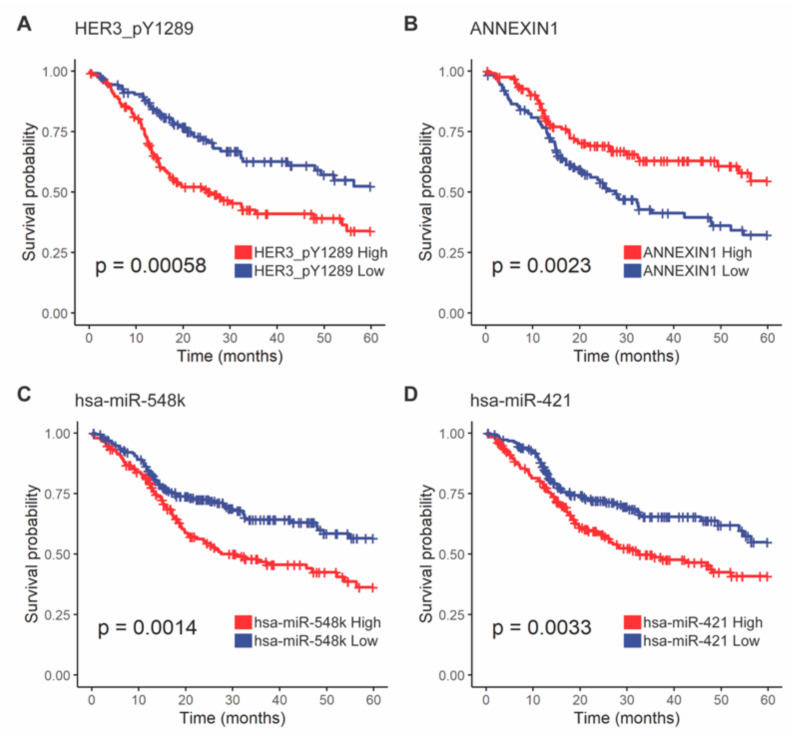
Overall survival differences by protein or miRNA abundance. Patients were grouped by (**A**) HER3_pY1289, (**B**) ANNEXIN1, (**C**) hsa-miR-538k, and (**D**) hsa-miR-421 abundance above or below the median. These proteins and miRNAs were significantly associated with overall survival with FDR correction using the Benjamini–Hochberg method. Log-rank tests were used for survival comparisons.

**Table 1 cancers-13-05313-t001:** Demographic differences by 3p arm deletion status in HPV-negative samples. Significant *p*-values are bolded.

		HPV-Negative Samples, No. (%) (*n* = 375)		
Variables		Threshold Arm Lost (*n* = 293)	Threshold Arm Preserved (*n* = 82)	*p*-Value
Age	Median (range)	61 (19–84)	64 (26–83)	0.053
Sex	Female	69 (24)	40 (49)	**<10^−4^**
	Male	224 (76)	42 (51)	
Anatomical site	Oropharynx	22 (8)	1 (1)	**<10^−3^**
	Hypopharynx	3 (1)	1 (1)	
	Larynx	82 (28)	8 (10)	
	Oral cavity	186 (63)	72 (88)	
Smoking history	Non-smoker	51 (23)	29 (41)	**<10^−3^**
	Light	29 (13)	14 (20)	
	Heavy	144 (64)	28 (39)	
T category	T0-T2/TX	112 (39)	35 (45)	0.30
	T3-T4	176 (61)	42 (55)	
N category	N0-N2a, NX	176 (61)	62 (81)	**0.0018**
	N2b-N3	111 (39)	15 (19)	
Overall stage	I–III	96 (37)	38 (51)	**0.033**
	IV	165 (63)	37 (49)	
Adjuvant radiotherapy	No	87 (34)	32 (42)	0.22
	Yes	170 (66)	44 (58)	
Distant metastasis	No	278 (95)	82 (100)	**0.049**
	Yes	15 (5)	0 (0)	

**Table 2 cancers-13-05313-t002:** Demographic differences by 3p arm deletion status in HPV-positive samples. Significant *p*-values are bolded.

		HPV-Positive Samples, No. (%) (*n* = 66)		
Variables		Threshold Arm Lost (*n* = 26)	Threshold Arm Preserved (*n* = 40)	*p*-Value
Age	Median (range)	59 (40–82)	56.5 (35–77)	0.22
Sex	Female	2 (8)	3 (8)	1
	Male	24 (92)	37 (92)	
Anatomical site	Oropharynx	17 (65)	31 (78)	0.52
	Hypopharynx	1 (4)	1 (2)	
	Larynx	1 (4)	0 (0)	
	Oral cavity	7 (27)	8 (20)	
Smoking history	Non-smoker	6 (26)	14 (41)	**0.015**
	Light	3 (13)	12 (35)	
	Heavy	14 (61)	8 (24)	
T category	T0-T2/TX	16 (73)	26 (76)	0.76
	T3-T4	6 (27)	8 (24)	
N category	N0-N2a, NX	17 (77)	21 (64)	0.38
	N2b-N3	5 (23)	12 (36)	
Overall stage	I–III	6 (40)	9 (38)	1
	IV	9 (60)	15 (62)	
Adjuvant radiotherapy	No	8 (36)	5 (14)	0.055
	Yes	14 (64)	32 (86)	
Distant metastasis	No	25 (96)	39 (98)	1
	Yes	1 (4)	1 (2)	

## Data Availability

Publicly available datasets were analyzed in this study. These data can be found here: https://www.cancer.gov/tcga (accessed on 15 June 2019). The validation dataset is available on request from the corresponding author, L.G.T.M. The analyzed data presented in this study are available within the main and Appendix A.

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
