# Peer review of "3p Arm Loss and Survival in Head and Neck Cancer: An Analysis of TCGA Dataset"

_cancers, 2021, doi:10.3390/cancers13215313_

Round 1

Reviewer 1 Report

The authors analyze TCGA data and compare tumors with and without 3p loss. The authors uncover some potentially interesting relationships (e.g. with ragnum score and T-cell status), though it is hard to know if 3p loss is causative (as implied throughout the text) and/or what the relationships are with PIK3CA and SOX2 on the other end of the chromosome. Ultimately, the authors present a re-analysis of TCGA data without any external datasets or deeper validation of findings. They also make several logical leaps in connecting one outcome to another (e.g. individual genes that might relate to 3p loss). The analysis would benefit from additional computational control experiments to help interpretation, or ideally from additional supporting data that extends beyond TCGA. Conceptually, validation of a role for 3p beyond the 3q oncogenes would be an advance, but I'm not convinced by the data shown by the authors that 3p is as important as its made out to be in the manuscript. 

Major points

1) Can the authors use additional publicly available data to confirm any of the relationships they have described? 

2) Many of these relationships have been described previously attributed to 3q gain, which makes the relationship between 3q gain and 3p loss more important to understand. The authors should re-analyzed data in the context of PIK3CA gain and/or mutation to see if 3p changes add to the differences previously described (or if they are independently associated). 

3) The analysis of RPPR data is not scientifically sound analysis and should be removed. Normally one would use the whole 3p-associated signature to test if that stratifies with survival, similar to what was found earlier with 3p loss, not cherry-pick candidates that are correlated with survival. 

4) Analysis of RNA signatures could have been better performed (especially using a signature of 3p gene expression). 

Author Response

1) Can the authors use additional publicly available data to confirm any of the relationships they have described?

We thank the reviewer for suggesting the critical value of a second independent dataset beyond the TCGA cohort (n=528). We agree, and we have included data from the MSK-IMPACT dataset (n=245) contributed by co-senior author Dr. Morris in the original submission. This second cohort adds significant value to the validity of our study beyond the earlier TCGA based analysis. We have adjusted the manuscript to make our analysis of the MSK-IMPACT cohort more visible throughout, as it provides important validation.

We have also added validation of the association of ANXA1 protein abundance with survival, using Clinical Proteomic Tumor Analysis Consortium data from HPV-negative head and neck cancer patients (n=110). This result can now be found in our manuscript (page 16).

2) Many of these relationships have been described previously attributed to 3q gain, which makes the relationship between 3q gain and 3p loss more important to understand. The authors should re-analyzed data in the context of PIK3CA gain and/or mutation to see if 3p changes add to the differences previously described (or if they are independently associated).

We thank the reviewer for this valuable suggestion. We have now controlled for PIK3CA gain in all instances where we used linear regression, including analyses of the immune and hypoxia environments, and RPPA. The findings can be seen in the revised manuscript on pages 13 to 16, and notably show that our key findings mostly remain significantly and independently associated with 3p arm loss, rather than PIK3CA gain. We have included these independent observations in our discussion.

3) The analysis of RPPR data is not scientifically sound analysis and should be removed. Normally one would use the whole 3p-associated signature to test if that stratifies with survival, similar to what was found earlier with 3p loss, not cherry-pick candidates that are correlated with survival.

We thank the reviewer for this comment but respectfully disagree about the approach we took to using the RPPA dataset. The organization of the RPPA screen represents a reproducible and methodical selection of proteins thought to be of biological importance.  Also, the analytic method that we used to analyze the RPPA data has been utilized in high-impact studies (i.e. 10.1038/ncomms4887). We have revised our RPPA analysis from Mann-Whitney U testing to linear regression with and without control for PIK3CA as requested in point 2 above. For survival analysis, we methodically selected for the top candidate changes that were significantly associated with 3p arm loss for further downstream survival analysis. Finally, we have added validation of our RPPA findings with an external dataset, as described in our response to point 1 above.

4) Analysis of RNA signatures could have been better performed (especially using a signature of 3p gene expression).

We thank the reviewer for this comment, which would improve our study. Unfortunately, we were unable to address this in our revisions within the allotted five days for resubmission. This is an important avenue for a future study.

Other comments:

We agree that there are other common mutations in HNSCC in addition to 3p arm loss, which may confound our analyses, and our findings explain associations rather than cause-and-effect relationships. We have adjusted the wording of our manuscript to make this more explicitly stated. We do offer possible hypotheses on the underlying biologic mechanisms and clinical significance in our discussion, based on review of the literature. However, we have aimed to disclose and acknowledge the limitations to studies like ours, which mine large databases to uncover a handful of molecular changes that warrant further investigation. Ultimately, we still feel that such studies add significant value to the existing literature and to guide further research.

Reviewer 2 Report

Authors demonstrated that 3p arm loss using new threshold can determine distinct molecular characteristics in HPV-negative patients, which may be clinically useful markers. Although authors analyzed several parameters using TCGA data based on their new classification, the impact is weak.

1) Why did authors give different threshold to define 3p arm deletion (HPV-negative patients are over 97% while HPV-positive group has over 50%). I assume all results presented here will depend on threshold.

2) Why can authors claim that alteration of several candidates (two proteins and miRNAs) depend on 3q status? Especially, region encoding miRNA-548k is near CCND1 and FADD at 11q13.3, which is commonly amplified in HNSCC.

3) As introduced by authors, Gross et al. previously performed multi-tiered genomic analysis where 3p loss is highlighted. What is novel point of this study? Did authors discover significant information? 

Author Response

1) Why did authors give different threshold to define 3p arm deletion (HPV-negative patients are over 97% while HPV-positive group has over 50%). I assume all results presented here will depend on threshold.

 We thank the reviewer for this comment. We had initially examined these thresholds based on our observation of the supplementary figure found in Gross et al.’s paper (PMID: 25086664), which showed different patterns of 3p arm loss between HPV-negative and HPV-positive samples. For instance, a visual observation of this supplementary figure shows that loss of the unstable 3p14.2 locus was an accurate surrogate for whole 3p arm loss in HPV-negative samples, but not for HPV-positive samples. This makes sense, given that HPV-positive tumors are known to be quite biologically distinct from HPV-negative tumors, and are thus treated as distinct biological entities in our study.

Indeed, when we visually plotted (Figure 1) the number of patients who had varying percentages of genes on the 3p arm deleted, we saw a convincing bimodal distribution of either <1% or >97% 3p arm loss in HPV-negative samples, but not in HPV-positive samples. Thus, only 48 of 423 (11%) patients were excluded in the HPV-negative cohort using the 97% threshold; whereas, 19 of 73 (26%) patients in the HPV-positive cohort would have been excluded using the same threshold, rather than the 5 of 73 (7%) patients we excluded using the 50% threshold. The resulting cohort had excellent concordance with the Gross et al. study, with 90% of patients with 3p14.2 locus loss and 0% of patients without 3p14.2 locus loss being included into our 3p arm loss group defined by the 50% deletion threshold. We have now further described this in our results (page 7).

2) Why can authors claim that alteration of several candidates (two proteins and miRNAs) depend on 3q status? Especially, region encoding miRNA-548k is near CCND1 and FADD at 11q13.3, which is commonly amplified in HNSCC.

 We thank the reviewer for this question. As with previous studies conducted at our lab, we have performed rigorous statistical analyses to report molecular findings that are more commonly found in samples with 3p arm loss than those without 3p arm loss, than would be expected by chance, with control for multiple testing. We agree that there are other common mutations in HNSCC in addition to 3p arm loss, which may confound our analyses, and our findings are indeed associative, not causative. In our discussion, it was not our intention to imply a causative effect of 3p arm loss, and we have adjusted the wording of our manuscript to make this more explicitly stated. We have aimed to disclose and acknowledge that this is a limitation to studies like ours which mine large databases to uncover a handful of molecular changes that warrant further investigation. However, we still feel that such studies add significant value to the existing literature, to guide further research that may include randomized controlled trials.

With regard to miRNA-548k in particular, we have mentioned in our discussion that the Gross et al. study (PMID: 25086664) also identified this transcript to be significantly altered in association with 3p arm status, similarly without making claims to a causative relationship. That we were able to reproduce this finding and discover other associations using our methodology supports the validity and value of our work, which provides numerous novel findings as outlined in the next response below.

3) As introduced by authors, Gross et al. previously performed multi-tiered genomic analysis where 3p loss is highlighted. What is novel point of this study? Did authors discover significant information?

We thank the reviewer for pointing out that Gross et al. in 2014 has previously conducted a multi-tiered analysis of molecular changes characterizing poor survival in association with 3p arm loss based on TCGA data. However, our study offers numerous novel and significant findings of value. Firstly, the aforementioned study used the data available in 2014 from a single TCGA cohort, which reflects 378 HNSCC patients, whereas we use the significantly expanded cohort of 528 patients. The purpose of the Gross et al. study was to scan for prognostic events across the TCGA head and neck cancer cohort, and indeed they identified and focused on the interaction between 3p arm loss and TP53 mutation, in addition to several other molecular changes including miR-548k. Our study, on the other hand, was conceptualized based on the existing knowledge that 3p arm loss is a possible prognostic marker that warrants further, more detailed exploration. This permitted us to focus specifically on elucidating a much more expansive array of molecular changes, encompassing more types of data and types of analysis than previously undertaken, that accompany the 3p arm loss event beyond TP53 mutation. We then methodically selected for the top candidate changes that were significantly associated with 3p arm loss for further downstream survival analysis. We have adjusted the manuscript to emphasize the significant and novel value of our findings in our introduction and discussion, which were resubmitted to the editor in a previous version without changes tracked.

Finally, we had included data from the MSK-IMPACT dataset contributed by co-senior author Dr. Morris in the original submission. This second cohort adds significant value to the validity of our study beyond the earlier TCGA based analysis. We have also adjusted the manuscript (in the previous version without changes tracked) to make our analysis of the MSK-IMPACT cohort more visible, as it provides important validation. We have also improved our manuscript with additional validation of RPPA data, and extended the significance by controlling our analyses for the commonly associated gain of PIK3CA on the 3q arm.

Reviewer 3 Report

In their study, the authors analyze the significance of p3 arm loss for survival in HNSCC, first defining the criteria (threshold definition) of p3 arm deletion status and determining the biological significance of this genomic event. Through a series of molecular analyzes, they find that regardless of the definition of p3 arm loss, it does not affect the survival of either HPV + or HPV tumors. However, according to the molecular-genetic profile, HPV- tumors with p3 arm loss proved to be a biologically distinct group of tumors with differences also in the tumor microenvironment and hypoxia level. The lower level of phospho-HER3 and ANXA1 proteins and the higher abundance of 59 miRNAs hsa-miR-548k and hsa-miR-421 correlated with survival. In the HPV + group of tumors, these differences were not observed.

The study is well designed and the methods used, including statistics, are appropriate. The Discussion is focused and concise and conclusions are supported with the results. The only major remark I have is that the authors do not report the results on progression-free survival that they otherwise predict in the Methods. Congratulations to the authors for their extensive but clearly presented work!

Minor comments:

Abstract, L57: abbreviation must be introduced before it is used (TME)

pg3/L121: What edition of the TNM staging system was used: 7th or 8th edition?

pg3/L119-120: 496 cases less than 85 years were analyzed (319 HPV+ and 73 HPV- cases); on pg5/L231-232, the numbers are 432 for HPW+ cases and 73 for HPV- cases.

The TCGA and MSK-IMPACT HNSCC cohorts should be described in term of treatment received (primary surgery vs. primary radio(chemo) therapy).

Table 1, table 2: What were the definitions of smoking history status (non-smoker, light, and heavy)? Was the first treatment in all patients surgery or were some of them primarily treated with definitive radio(chemo)therapy?

pg7/L276: advanced nodal disease = N2bN-N3 N-category

pg7/L277: advanced overall stage = stage IV

pg8-9/L286-305 and pg15-16/L400-432 (Fig.7) : Results of the progression-free survival analysis should be presented (as announced in the Material and Methods section, pg3/L137).

Author Response

The reviewer liked our manuscript and offered praise that “the study is well designed and the methods used, including statistics, are appropriate. The Discussion is focused and concise and conclusions are supported with the results.” They also felt that the work was “extensive but clearly presented”. We thank the reviewer for their kind comments. Minor revisions as suggested can now be found within the manuscript (present in the version previously resubmitted to the editor without changes tracked).

Round 2

Reviewer 1 Report

The authors have addressed some of my concerns appropriately, especially through the additional of validation data sets. 

Reviewer 2 Report

I have no comments.